# Modeling osteoporosis to design and optimize pharmacological therapies comprising multiple drug types

**David J Jörg[1]\*, Doris H Fuertinger[1], Alhaji Cherif[2], David A Bushinsky[3], Ariella Mermelstein[2], Jochen G Raimann[2], Peter Kotanko[2,4]**

[1]Biomedical Modeling and Simulation Group, Global Research and Development, Fresenius Medical Care Germany, Bad Homburg, Germany; [2]Renal Research Institute, New York, United States; [3]Department of Medicine, University of Rochester School of Medicine and Dentistry, Rochester, United States; [4]Icahn School of Medicine at Mount Sinai, New York, United States

**Abstract** For the treatment of postmenopausal osteoporosis, several drug classes with different mechanisms of action are available. Since only a limited set of dosing regimens and drug combinations can be tested in clinical trials, it is currently unclear whether common medication strategies achieve optimal bone mineral density gains or are outperformed by alternative dosing schemes and combination therapies that have not been explored so far. Here, we develop a mathematical framework of drug interventions for postmenopausal osteoporosis that unifies fundamental mechanisms of bone remodeling and the mechanisms of action of four drug classes: bisphosphonates, parathyroid hormone analogs, sclerostin inhibitors, and receptor activator of NF-κB ligand inhibitors. Using data from several clinical trials, we calibrate and validate the model, demonstrating its predictive capacity for complex medication scenarios, including sequential and parallel drug combinations. Via simulations, we reveal that there is a large potential to improve gains in bone mineral density by exploiting synergistic interactions between different drug classes, without increasing the total amount of drug administered.

**\*For correspondence:**
david.joerg@fmc-ag.com

## Editor's evaluation

The authors have developed a mathematical framework of drug interventions for postmenopausal osteoporosis using bisphosphonates, parathyroid hormone analogs, romosozumab, and denosumab. After calibrating and validating the model, authors demonstrated a predictive ability for complex clinical scenarios including sequential and parallel drug combinations. These data may be of great help in clinical practice.

## Introduction

Osteoporosis, a disease characterized by porous bone prone to fractures, affects hundreds of millions of people worldwide (*Cooper and Ferrari, 2019*; *Hernlund et al., 2013*). Most recent estimates place the global annual incidence of bone fragility fractures at 9 million in the year 2000 (*Cooper and Ferrari, 2019*); projections for the year 2050 suggest between 7 and 21 million annual hip fractures (*Gullberg et al., 1997*). Osteoporosis-associated bone fractures lead to disabilities, pain, and increased mortality (*Cooper and Ferrari, 2019*). In the United States, medical cost for osteoporosis, including inpatient, outpatient, and long-term care costs, has been estimated at US\$17 billion in 2005 (*Burge et al., 2007*); in the European Union, the total cost of osteoporosis, including pharmacological

**eLife digest** Our bones are constantly being renewed in a fine-tuned cycle of destruction and formation that helps keep them healthy and strong. However, this process can become imbalanced and lead to osteoporosis, where the bones are weakened and have a high risk of fracturing. This is particularly common post-menopause, with one in three women over the age of 50 experiencing a broken bone due to osteoporosis.

There are several drug types available for treating osteoporosis, which work in different ways to strengthen bones. These drugs can be taken individually or combined, meaning that a huge number of drug combinations and treatment strategies are theoretically possible. However, it is not practical to test the effectiveness of all of these options in human trials. This could mean that patients are not getting the maximum potential benefit from the drugs available.

Jörg et al. developed a mathematical model to predict how different osteoporosis drugs affect the process of bone renewal in the human body. The model could then simulate the effect of changing the order in which the therapies were taken, which showed that the sequence had a considerable impact on the efficacy of the treatment. This occurs because different drugs can interact with each other, leading to an improved outcome when they work in the right order.

These results suggest that people with osteoporosis may benefit from altered treatment schemes without changing the type or amount of medication taken. The model could suggest new treatment combinations that reduce the risk of bone fracture, potentially even developing personalised plans for individual patients based on routine clinical measurements in response to different drugs.

interventions and loss of quality-adjusted life-years (QALYs), is projected to rise from about €100 billion in 2010 to €120 billion in 2025 (*Odén et al., 2013*).

Osteoporotic bone is the consequence of an imbalance of continuous bone resorption and bone formation, which—under close to homeostatic conditions—has the function to remove microfractures and renew the structural integrity of bone. Postmenopausal women are particularly at risk of osteoporosis: the rapid decline of systemic estrogen levels after menopause and other aging-related effects such as increased oxidative stress contribute to or drive the development of osteoporosis (*Riggs et al., 1998*; *Manolagas, 2010*). Moreover, osteoporosis can be a sequela of diseases affecting bone metabolism and remodeling such as primary hyperparathyroidism or gastrointestinal diseases (*Painter et al., 2006*). Osteoporosis can also be a side effect of treatments for other diseases; as a prime example, glucocorticoid administration is the most common cause of secondary osteoporosis (*Weinstein, 2012*). Over the last decades, an array of different osteoporosis treatments have emerged, from simple dietary supplementations such as calcium and vitamin D to specialized drugs targeting bone-forming and -resorbing cells and related signaling pathways (*Tu et al., 2018*). This entails a plethora of different medication options, including a large number of possible dosing schemes and combinations of drugs, administered in sequence or in parallel. Due to the huge number of such treatment schemes and the required time from study inception to completion, very few of them have been clinically tested so far when compared to the total number of available options.

Concomitant with the development of new osteoporosis drugs, mathematical and biophysical modeling approaches capturing bone-related physiology have advanced our quantitative understanding of the biological principles governing bone mineral metabolism, bone turnover, and development of osteoporosis. Pioneering work by *Lemaire et al., 2004* describes the dynamics of bone-forming and -resorbing cell populations coupled through signaling pathways and could qualitatively reproduce the effects of senescence, glucocorticoid excess, and estrogen and vitamin D deficiency on bone turnover. Since then, compartment-based descriptions of the mineral metabolism, bone-forming and -resorbing cell populations, and related signaling factors have elucidated the role of essential regulatory mechanisms underlying mineral balance and bone turnover (*Komarova et al., 2003*; *Lemaire et al., 2004*; *Pivonka et al., 2008*; *Pivonka et al., 2010*; *Peterson and Riggs, 2010*; *Zumsande et al., 2011*; *Schmidt et al., 2011*; *Graham et al., 2013*; *Tanaka et al., 2014*; *Komarova et al., 2015*; *Berkhout et al., 2015*). Coarse-grained as well as detailed spatially extended descriptions of bone geometry have also addressed the effects of mechanical forces and the propagation of the multicellular units responsible for bone turnover (*Ryser et al.,*

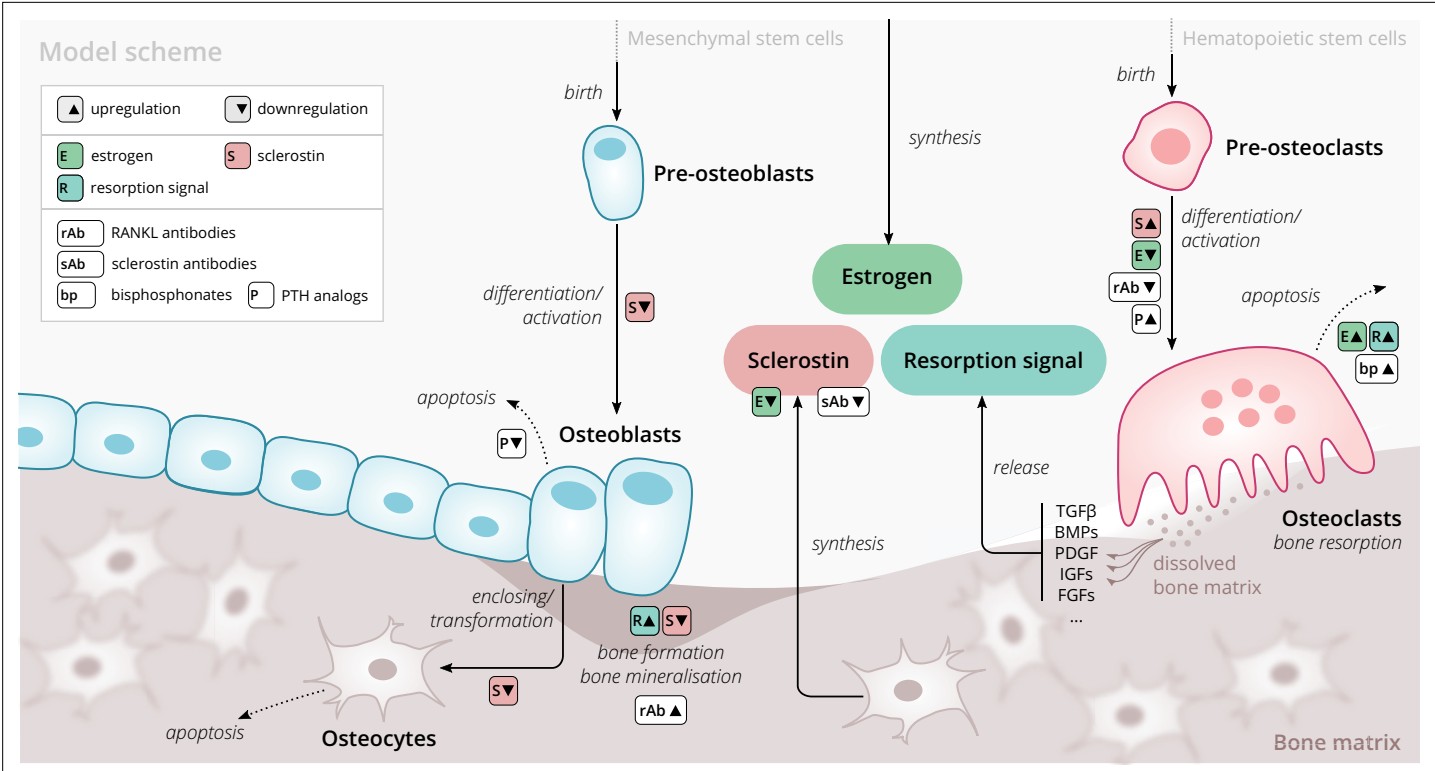

**Figure 1.** Schematic of the osteoporosis model describing the cell dynamics and signaling pathways within a 'representative bone remodeling unit (BRU)'. Regulatory interactions between different model components are indicated by colored boxes (see legend). TGFβ, transforming growth factor beta; BMP, bone morphogenetic protein; PDGF, platelet-derived growth factor; IGF, insulin-like growth factor; FGF, fibroblast growth factor.

*2009*; *Buenzli et al., 2011*; *Scheiner et al., 2013*; *Buenzli et al., 2014*; *Pivonka et al., 2013*), as well as the influence of secondary diseases such as multiple myeloma (*Ayati et al., 2010*). Detailed models of bone remodeling and calcium homeostasis have become versatile and widely used tools in hypothesis testing, such as the seminal model by *Peterson and Riggs, 2010*, which includes submodels for various organs such as gut, kidney, and the parathyroid gland. Pharmacokinetic and pharmacodynamic (PK/PD) models of therapeutic interventions have mostly focused on capturing the mechanisms of action of a single or a few drugs and testing their dosing regimens (*Marathe et al., 2008*; *Marathe et al., 2011*; *Ross et al., 2012*; *Scheiner et al., 2014*; *Eudy et al., 2015*; *Lisberg et al., 2017*; *Martínez-Reina and Pivonka, 2019*; *Zhang and Mager, 2019*). Recent modeling efforts have also started addressing the effects of drug combinations on bone-forming and -resorbing cells, pointing out the need for corresponding model frameworks to include clinically relevant variables like bone mineral density (BMD) and bone turnover biomarkers (BTMs) (*Lemaire and Cox, 2019*), as well as combination therapies of physical exercise and drug treatment (*Lavaill et al., 2020*). An integrated mathematical framework for multiple drugs, which can also be used to quantitatively predict the effects of drug combinations in sequence and in parallel is not yet available.

Building on established mechanisms of bone turnover, we here present a quantitative model of bone turnover and postmenopausal osteoporosis treatment, unifying the description of multiple classes of drugs with different mechanisms of action, namely, bisphosphonates, parathyroid hormone (PTH) analogs, sclerostin antibodies, and receptor activator of NF-κB ligand (RANKL) antibodies. We calibrate the model using published population-level data from several clinical trials and assess its ability to predict the outcome of previously conducted clinical studies based on the medication scheme alone. We then use the model to demonstrate how medication schemes involving drug combinations can be optimized for a given medication load and discuss future model extensions.

## Mechanisms of bone turnover and its regulation

Our model is based on a small set of key principles of bone turnover, which we briefly recapitulate here (*Figure 1*). As a composite tissue comprising hydroxyapatite, collagen, other proteins, and water (*Boskey, 2013*), bone is constantly turned over to renew its integrity and remove microdamage, at an average rate of about 4% per year in cortical bone and about 30% per year in trabecular bone (*Manolagas, 2000*).

### Bone-resorbing and -forming cells

Bone resorption is performed by osteoclasts, multinucleated cells formed through the differentiation and fusion of their immediate precursors (pre-osteoclasts), which are derived from pluripotent hematopoietic stem cells via the myeloid lineage (*Boyce and Xing, 2008*). Osteoclasts attach to bone tissue and resorb it through the secretion of hydrogen ions and bone-degrading enzymes (*Fuller and Chambers, 1995*), which leads to the release of minerals and signaling factors stored in the bone matrix. New bone is formed by osteoblasts, a cell type derived from mesenchymal stem cells via several intermediate states that give rise to pre-osteoblasts and finally osteoblasts (*Eriksen, 2010*). Groups of osteoblasts organize into cell clusters (osteons) and collectively lay down an organic matrix (osteoid), which subsequently becomes mineralized over the course of months. Osteoblasts that are enclosed in the newly secreted bone matrix become osteocytes, nondividing cells with an average life span of up to several decades. Osteoclasts and osteoblasts organize into spatially defined local clusters termed 'bone remodeling units' (BRUs) (*Figure 1*), in which osteoblasts replenish the bone matrix previously resorbed by osteoclasts with a delay of several weeks. In cortical bone, the outer protective bone layer, BRUs migrate as a whole in 'tunnels,' whereas within the inner cancellous bone, BRUs propagate on the surfaces of the trabeculae, renewing the bone matrix in the process (*Eriksen, 2010*).

### Signaling pathways

The differentiation and activity of osteoclasts and osteoblasts are regulated through several signaling pathways and hormones; recent reviews provide comprehensive descriptions of the various pathways (*Siddiqui and Partridge, 2016*). Osteoclast formation and activity are prominently regulated by RANKL and macrophage colony-stimulating factor (M-CSF) synthesized by bone marrow stromal cells. RANKL binds to receptor activator of NF-$\kappa$B (RANK) on osteoclast precursors and promotes their differentiation into mature osteoclasts; osteoprotegerin (OPG) acts as a decoy receptor for RANKL and thus inhibits bone resorption (*Boyce and Xing, 2008*; *Clarke, 2008*). When laying down new bone, osteoblasts store signaling factors in the bone matrix, including transforming growth factor beta (TGFβ), bone morphogenetic protein (BMP), insulin growth factors (IGFs), platelet-derived growth factor (PDGF), and fibroblast growth factors (FGFs) (*Solheim, 1998*). Upon bone resorption, these factors are released and regulate cell fates and activity of osteoblasts and osteoclasts, thereby coupling bone resorption and formation (*Houde et al., 2009*; *Eriksen, 2010*). Osteocytes secrete sclerostin, a Wnt inhibitor interfering with extracellular binding of Wnt ligands (*Li et al., 2005*). Sclerostin inhibits bone formation and promotes resorption via downregulation of osteoblastogenesis and upregulation of osteoclastogenesis (*Delgado-Calle et al., 2017*; *Maré et al., 2020*). Since bone also acts as a mineral reservoir for the body, regulators of calcium homeostasis such as PTH and vitamin D also strongly affect the balance of bone formation and resorption alongside the intestinal absorption and renal reabsorption of calcium (*Mundy and Guise, 1999*).

### Estrogen

The sex hormone estrogen inhibits bone resorption by inducing apoptosis of osteoclasts (*Kameda et al., 1997*) and lowering circulating sclerostin levels (*Mödder et al., 2011*). The rapid decline of estrogen levels after menopause is one known cause of postmenopausal osteoporosis (*Riggs et al., 1998*).

## Results

### Model overview

The primary purpose of our model is to provide an efficient representation of bone turnover on multiple time scales from weeks to decades that allows for the quantitative description of drug interventions.

Of particular interest are the consequences of pharmacological therapies on long-term dynamics of the BMD in specific bone sites and biochemical markers of bone formation and resorption. To this end, we considered a minimal set of physiologically relevant dynamic components (*Figure 1*) that are sufficient to capture a large range of clinically observed population-level data on drug interventions. Thus, our model describes a 'representative BRU' that abstracts from the vast set of intricate regulatory mechanisms underlying calcium homeostasis or the complex bone geometry.

Our model comprises the following dynamic components to describe the bone turnover through a representative BRU: cell densities of (i) pre-osteoclasts, (ii) osteoclasts, (iii) pre-osteoblasts, (iv) osteoblasts, (v) osteocytes, (vi) sclerostin concentration, (vii) total bone density, and (viii) bone mineral content (BMC). The BMD is given by the product of bone density and BMC. Osteoblasts and osteoclasts can undergo apoptosis and are derived from pre-osteoblasts and pre-osteoclasts, respectively, with differentiation rates that depend on regulatory factors such as estrogen and sclerostin (*Figure 1*). Pre-osteoblasts and pre-osteoclasts are formed at constant rates and undergo apoptosis. These progenitor populations provide a dynamic reservoir for rapid differentiation and activation of osteoblasts and osteoclasts, respectively, which can be temporarily depleted if stimulated by a drug intervention. Osteocytes are derived from osteoblasts and provide a source of sclerostin, which has a regulatory effect on osteoblasts, osteoclasts, and thus, bone density change. The gain and loss rates of bone density are proportional to the density of osteoblasts and osteoclasts, respectively. The BMC has a steady state whose level can be temporarily shifted through drug administration, effectively accounting for more complex underlying dynamics such as promotion of secondary mineralization. All rates of cell formation, differentiation, apoptosis, and bone formation and resorption generally depend on the concentration of sclerostin, estrogen, and a 'resorption signal.' These dependencies also implicitly account for regulation of bone remodeling via other routes, for example, the RANK–RANKL–OPG pathway. The effects of aging and the onset of menopause are represented through an age-dependent serum estrogen concentration, which has been determined from the literature (*Sowers et al., 2008*; Appendix 1). The resorption signal corresponds to the melange of signaling factors stored in the bone matrix. Therefore, its release is proportional to the rate of bone resorption. The serum concentration of BTMs such as the resorption marker C-terminal telopeptide (CTX), the formation markers procollagen type 1 amino-terminal propeptide (P1NP), and bone-specific alkaline phosphatase (BSAP) were identified with elementary functions of the bone resorption and formation rates in the model (Appendix 1).

We extended this core model of long-term bone turnover by a dynamic description of the mechanisms of action of several drug classes used in osteoporosis treatment: RANKL antibodies (denosumab), sclerostin antibodies (romosozumab), bisphosphonates (alendronate and others), and PTH analogs (teriparatide) (Appendix 2). We also included blosozumab, another sclerostin inhibitor, which was investigated in osteoporosis trials but not approved for osteoporosis treatment at the time the present work was conducted. PTH is known to exert anabolic or catabolic effects depending on whether administration is intermittent or continuous (*Tam et al., 1982*; *Hock and Gera, 1992*); PTH description in our model is restricted to the anabolic administration regimes relevant for osteoporosis treatment. A schematic overview of all model components, mechanisms, and regulatory interactions is given in *Figure 1*; a detailed formal description of the model and its extensions is provided in Appendix 1 and Appendix 2.

## Capturing clinical study results with the model

The model and the corresponding medication modules rely on an array of physiological parameters (rates of cell formation, differentiation and death, concentration thresholds for signaling activity, medication efficacies and half-lives, etc.) many of which are not directly measureable. However, clinical measurements on physiological responses to medications with different mechanisms of action provide a wealth of indirect information about time scales of bone turnover and regulatory feedbacks. We calibrated the model using published clinical data from various seminal studies on both (i) long-term BMD age dependence and (ii) the response of BMD and BTMs to the administration of different drugs (see *Appendix 3—table 2* for a comprehensive list of data sources). Although BMD constitutes the major target variable of our model, the dynamics of BTM concentrations carry important complementary information about the mode of action of the administered drugs (antiresorptive, anabolic, and combinations) that crucially informs the calibration procedure. To allow the model to capture the

effects of medications as physiologically sensible modulations of the age-dependent bone mineral metabolism, we created hybrid datasets each of which comprised both aging-related BMD changes and the response to a treatment (see 'Methods' and *Appendix 1—figure 1D*).

We then determined a single set of model parameters through a simultaneous fit of the free 31 model parameters to capture a specified set of hybrid aging/treatment datasets containing different drug responses (Appendix 3). Without constraining the average rate of skeletal bone turnover, model calibration yielded an inferred value of about 6% per year on average, of the same order of values reported for cortical bone, which constitutes about 75% of the skeleton (*Manolagas, 2000*). The model was able to capture the BMD and BTM dynamics across all calibration datasets with remarkable accuracy (*Appendix 3—figure 1*), despite the model's structural simplicity. To quantify the goodness of the fit, we computed the mean absolute percentage error (MAPE) between model simulations and clinical data; the MAPE for BMD was consistently below 1% for all calibration datasets (*Appendix 3— table 3*), indicating an excellent agreement between model and data. The qualitative behavior of BTMs (i.e., the direction of their excursions from baseline) was captured correctly in all calibration datasets, indicating an adequate description of the drugs' mode of action in the model; relative deviations in the total magnitude of BTM excursions observed for some datasets were mostly due to slight offsets in the timing of peaks and troughs and low absolute values of the respective BTM concentrations, as highlighted by comparing different goodness measures (Appendix 3 and *Appendix 3—table 3*).

After obtaining the reference parameter set, we sought to validate the calibrated model by assessing its ability to predict the effects of drug dosing schemes that had not been used for calibration. Model validation included complex sequential and parallel drug combinations and therefore challenged the model to predict the effects of treatment schemes beyond those used in calibration (*Appendix 3— table 2*). To this end, the model received only drug dosing information used in the respective clinical trials but was not informed by BMD or BTM measurements, which instead it had to predict. With the single set of previously determined parameters, the model showed a remarkable capacity to quantitatively forecast the effects of a multitude of medication schemes, both during treatment and follow-up periods (*Figure 2*, *Figure 2—figure supplement 1*). Even in scenarios including sequential treatments with up to three different drug types and parallel treatments with two different drugs, respectively, the model was able to predict the complex progression of both BMD and biomarker levels with a high degree of accuracy (*Figure 2*). Across all validation datasets, MAPEs for BMD were consistently below 1.5% (*Appendix 3—table 3*), indicating an excellent predictive capacity of the model. In summary, this validation provided a strong corroboration of the model's capacity to capture the physiological dynamics of bone turnover and the mechanisms of action of various drugs relevant to osteoporosis treatment using a single set of model parameters.

## Testing alternative treatment schemes

Having established the predictive capacity of the model for the considered medications, we aimed to utilize the model to study and optimize hypothetic drug dosing regimens. As an example, we considered a sequential treatment with three drugs of different types: the bisphosphonate alendronate, the sclerostin inhibitor romosozumab, and the RANKL inhibitor denosumab. In a clinical trial reported by *McClung et al., 2018*, the sequence alendronate (70 mg per week for 1 year), followed by romosozumab (140 mg per month for 1 year), followed by denosumab (60 mg per 6 months for 1 year) had been studied (*Figure 2*). However, in principle there are six different sequences in which these drugs can be administered: ARD, ADR, DAR, DRA, RAD, and RDA (A: alendronate; R: romosozumab; D: denosumab). A priori, it is not obvious whether synergistic or antagonistic interactions between these drugs and the physiological state in which they leave the patient may lead to a differential short- and long-term evolution of BMD and biomarkers between different medication sequences. Probing all six sequences in a clinical trial would present a time- and resource-consuming endeavor and inevitably expose part of the study population to suboptimal treatment schemes. Instead, we probed these different treatment options using the present model (*Figure 3A*). To assess the predicted clinical success of different sequences, we compared two clinically relevant outcomes across different schemes: (i) the maximum achieved BMD increase (as compared to baseline at treatment start) irrespective of when it occurred (*Figure 3B*) and (ii) the residual long-term effects of treatment on BMD as monitored by the relative BMD 10 years after treatment end (*Figure 3C*) .

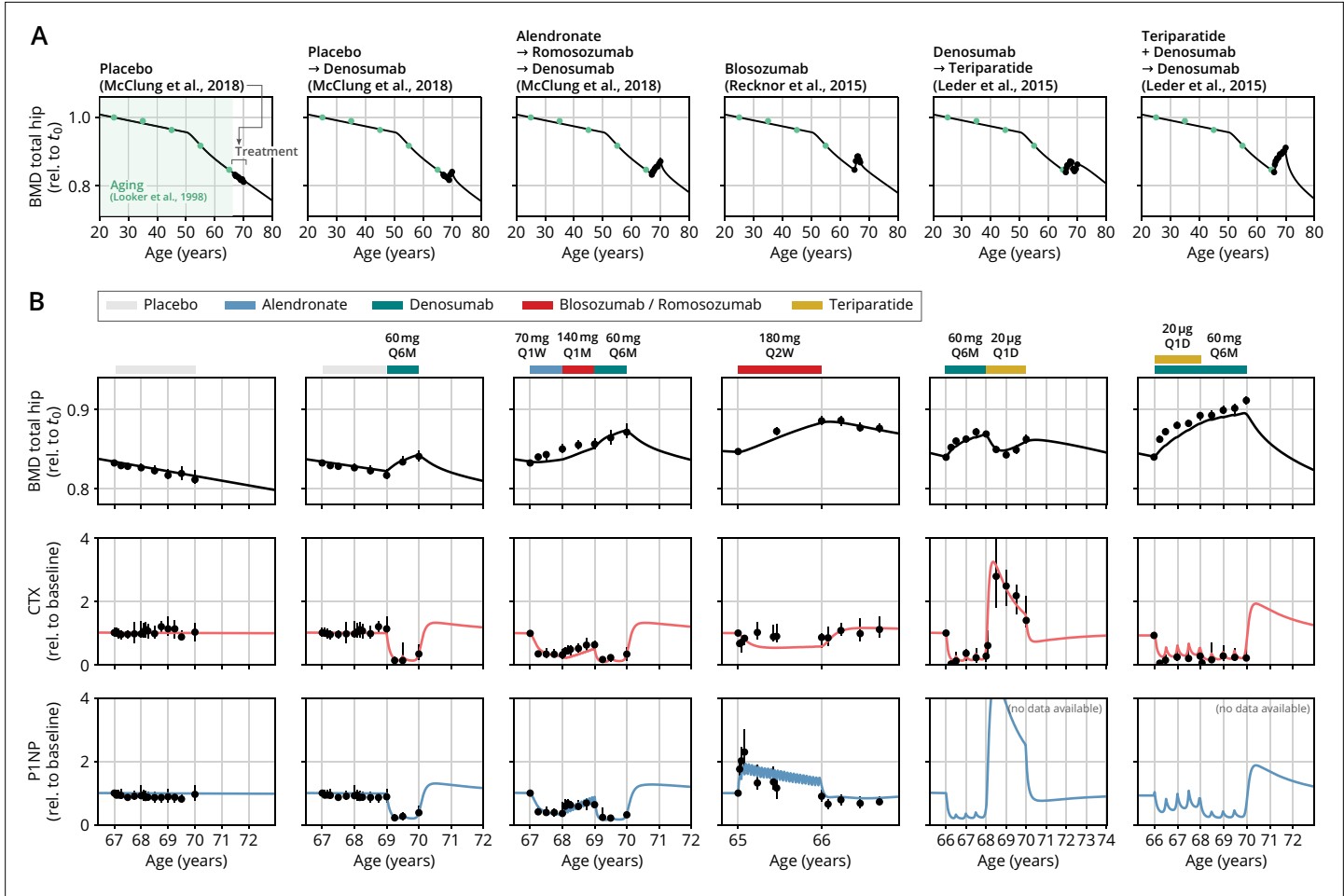

**Figure 2.** With a single set of parameters, the calibrated model can quantitatively predict the effects of various drugs in different dosing regimens, alone and in combination. (**A**) Comparison of simulated total hip bone mineral density (BMD, black curves) and clinical data (dots), including aging behavior (green dots) and treatment behavior (black dots) of various sequential drug treatments, including denosumab, romosozumab, alendronate, and teriparatide. Hybrid aging/treatment datasets were created combining data from *Looker et al., 1998* (aging dataset, green dots in panel **A**; in total $N = 3251$ subjects 20 years and older), as well as *Recknor et al., 2015* (blosozumab 180 mg Q2W: $N = 25$), *McClung et al., 2018* (placebo/deno.: $N = 18$, alendro./romo./deno.: $N = 21$), and *Leder et al., 2015* (deno./teri.: $N = 27$, teri. + deno./deno.: $N = 23$) (treatment datasets, black dots in panels **A** and **B**) as indicated, see 'Methods.' (**B**) Zoom into the treatment regions shown in panel (**A**) including BMD (black) and baseline changes of the bone resorption marker C-terminal telopeptide (CTX, red) and the bone formation marker procollagen type 1 amino-terminal propeptide (P1NP, blue). Colored bars above the plots indicate the medication scheme (see legend). Data points show population averages; average types and error bar types as reported in the respective original publication. In both panels, BMD is displayed as a fraction of its value at $t_0 = 25$ years.

The online version of this article includes the following figure supplement(s) for figure 2:

**Figure supplement 1.** Continuation of *Figure 2* comparing model predictions and clinical data from various studies, all conventions identical.

Indeed, we found that the outcomes of different medication sequences were markedly different despite the same total amount of drug administered (*Figure 3A*). Some sequences (such as ARD and RAD) reached a considerably higher maximum BMD during the course of the simulated treatment, which allowed us to rank treatments according to maximum BMD gain (*Figure 3B*). Notably, while some sequences were superior to others as measured by the maximum BMD increase during treatment, they performed markedly worse (as compared to, e.g., DRA and RDA) with regard to long-term BMD evolution as predicted by model simulations (*Figure 3C*). This behavior suggests that short-term BMD gains may be limited as a proxy for the clinical benefit of a treatment as a whole. Within our modeling scheme, the explanation for this behavior is found in differing 'rebound' effects after treatment end: simulated drug-mediated inhibition of osteoclastogenesis leads to a build-up of an undifferentiated osteoclast precursor pool. After treatment end, this precursor pool becomes licensed to differentiate and rapidly gives rise to a large active osteoclast population, leading to accelerated

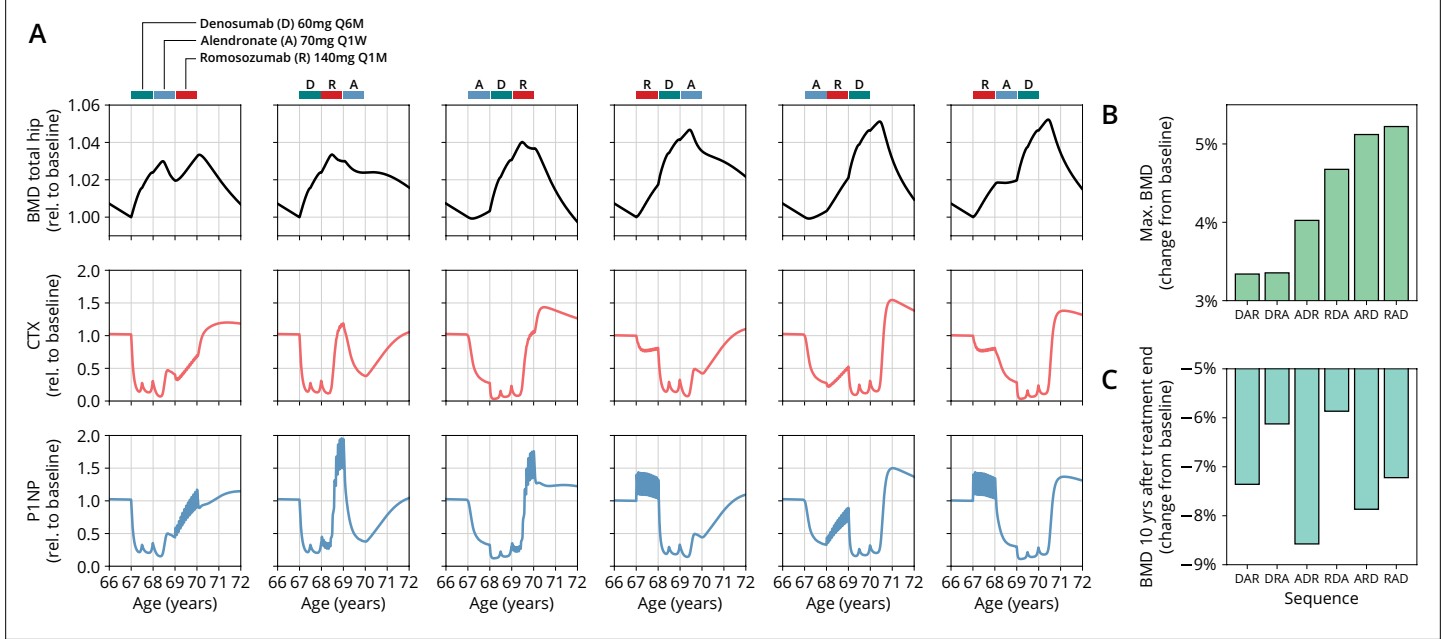

**Figure 3.** The model predicts differential outcomes for different sequences of the same drugs at constant total medication load. (**A**) Simulated progression of bone mineral density (BMD) and C-terminal telopeptide (CTX) and procollagen type 1 amino-terminal propeptide (P1NP) concentrations for different sequences (columns) of the three drugs denosumab (D), alendronate (A), and romosozumab (R) as indicated. Simulated treatment starts at age 67. The total amount of drug administered is identical among columns. Clinical results on the sequence ARD (column 5) were reported in *McClung et al., 2018*, see also *Figure 2*. (**B**) Maximum simulated BMD (relative to baseline at treatment start) achieved during the course of treatment for different drug sequences. (**C**) Simulated BMD 10 years after treatment end (relative to baseline at treatment start) for different drug sequences.

resorption of the bone matrix that had been built up during treatment. In this paradigm, specific drug sequences lead to an attenuation of this effect, for example, by enhancing osteoclast apoptosis during such a 'rebound' phase, thereby modulating bone turnover in the long run.

In summary, our model analysis suggests considerable potential in the improvement of dosing regimens and drug sequencing in osteoporosis treatment, especially combination therapies, to achieve an optimal effect for a given medication load. These improvements are possible because the mechanisms of action of one drug may act either favorably or adversely on the state of the bone mineral metabolism left behind by the preceding treatment with another drug.

## Discussion

Treatment of osteoporosis is complex, expensive, and in many circumstances opinion-based. With bone physiology as our guiding principle, we have introduced a mathematical modeling framework that can quantitatively capture and predict the progression of osteoporosis in postmenopausal women with and without medical therapy. Our model is built on a small set of essential mechanisms of bone turnover. The effectivity of this approach suggests that—despite the complexity of the bone mineral metabolism—the dynamics relevant for osteoporosis medications can be condensed into only a few components. These components describe the biology of osteoblasts, osteoclasts, and osteocytes, as well as their precursor cell populations and a few essential regulatory feedbacks through hormones and signaling factors such as estrogen, sclerostin, and bone-matrix-derived factors.

The general nature of the model allowed us to capture the BMD and BTMs of a multitude of clinical treatment studies. Notably, the model can also predict the effects of a broad range of drug dosing regimens and complex drug combination therapies beyond those used for model development. This corroborates the model's predictive capacity, supporting its use for the design of future clinical trials. However, it is important to note that some parameters (e.g., concentration thresholds for signaling factors) were inferred through the model calibration procedure from BMD and BTM dynamics alone. Hence, when used as a predictive tool, general quantitative limitations of the model have to be

considered, especially when extrapolating into extreme dosing regimens, dosing frequencies, or age regions beyond the validated ones.

It is of clinical relevance that exemplary model predictions suggest a large potential for the development of optimized combination therapies involving different drug types and treatment schemes. These may range from a simple rearrangement of a sequence of drugs at given total drug doses (as shown in this article) to complex interwoven or cyclic administration schemes that exploit synergistic effects between different medication types. Notably, model simulations extrapolating the long-term BMD development after treatment end suggest that medication schemes eliciting a rapid BMD increase are not necessarily accompanied by a sustained elevation of the BMD. Instead, some initially successful treatment schemes may lead to a 'rebound' effect of accelerated bone loss after treatment end, a prediction that cautions against using short-term BMD increases as the sole proxy for treatment success. Such extrapolations into follow-up periods long after treatment end, which are mostly inaccessible to clinical studies, highlight the potential role of the model in considering long-term treatment success when optimizing treatment schemes.

In our research, we have focused on postmenopausal osteoporosis, the most widespread type of osteoporosis. However, the generic manner in which the model represents bone remodeling and the effects of medications renders it a general platform for the study of treatments that can be adapted to other types of primary and secondary osteoporosis. The modular nature of the model enables future extensions; besides additional medication types, these may include the effects of comorbidities that elicit osteoporosis or interact with it (such as primary and secondary hyperparathyroidism), medications that contribute to osteoporosis (such as glucocorticoid therapy), lifestyle-dependent factors such as smoking and alcohol consumption, the effects of dietary supplementation of osteoporosis treatment through calcium and vitamin D and effects of microgravity on bone, as experienced by astronauts on extended missions in space. Physical activity is another important contributor to bone remodeling, which we have not considered here. Detailed modeling approaches involving biomechanical feedback suggest synergistic effects between drug treatment of osteoporosis and physical activity (*Lavaill et al., 2020*). Such results call for a further exploration of integrated approaches to osteoporosis therapy combining pharmacological treatment and lifestyle adjustments.

Clearly, the goal of osteoporosis therapy is the reduction of fracture risk during and after therapy. While BMD has a prime role in the evaluation of osteoporosis therapies and can be measured rather easily using dual-energy x-ray absorptiometry (DXA), its relationship to fracture risk is complex. Fracture risk calculations used in clinical practice also involve demographic and lifestyle-related factors while mostly relying on BMD point measurements (*Kanis et al., 2009*). However, the quantitative associations between BMD, age, and fracture risk reported in many studies (*Kanis et al., 2001*; *Berger et al., 2009*; *Austin et al., 2012*; *Krege et al., 2013*; *Black et al., 2018*; *Ensrud et al., 2022*) can be used to create statistical models that may relate entire BMD time courses to a patient's fracture risk. Combining such statistical models with the physiology-based model presented here would enable to optimize therapies directly for a minimized long-term fracture risk instead of maximized BMD gain. Thus, our model can serve as a quantitative starting point for the forecast of pharmacological therapies of osteoporosis but also highlights the role of mechanistic mathematical descriptions in understanding the biological principles of drug action.

## Methods
### Hybrid aging/treatment datasets
To create hybrid aging/treatment datasets, we merged a dataset comprising the BMD age dependence from *Looker et al., 1998* with different clinical study datasets containing the BMD response to various medications (*Appendix 3—table 2*). The aging dataset from *Looker et al., 1998* consisted of mean total femur BMD measurements in non-Hispanic white, non-Hispanic black, and Mexican American women, reported in 10-year age bins ranging from 20 to 80 years and older. We used bin averages as proxy BMD indicators for the center of the respective age window (*Appendix 1—figure 1B*). Rescaling the reported means for the three ethnic groups to their value for the earliest age bin revealed that relative changes in BMD were remarkably consistent among ethnic groups (*Appendix 1—figure 1C*) despite differing absolute baselines. Therefore, and since the model only addresses relative BMD

changes, we resorted to the dataset with the largest underlying study population for calibration, which was the dataset comprising the non-Hispanic white female study population. Datasets on the response to medications from clinical trials on romosozumab, blosozumab, denosumab, alendronate, and teriparatide consisted of study population averages of total hip BMD and serum concentrations of one or more BTMs (CTX, P1NP, BSAP) during the treatment, and if available, during a follow-up period. Reported study population averages on the respective quantities were digitized directly from the data figures in the corresponding publications (*Appendix 3—table 2*).

To merge aging and treatment datasets, the BMD from treatment datasets was rescaled such that the BMD baseline at treatment start corresponds to the linearly interpolated age-dependent BMD at treatment start. The treatment start was placed at the average age of the study population upon study start (rounded to full years) as reported in the respective publication (*Appendix 1—figure 1D*). BTM measurements were normalized to baseline values.

## Acknowledgements

We thank Friederike E Thomasius for insightful discussions and critical comments on the manuscript.

## Additional information

### Competing interests

David J Jörg: DJJ is an employee of Fresenius Medical Care Germany. DJJ is an inventor on a patent application named "Virtual Osteoporosis Clinic" (WO 2021/231374 A1). The author has no other competing interests to declare. Doris H Fuertinger: DHF is an employee of Fresenius Medical Care Germany. DHF is an inventor on a patent application named "Virtual Osteoporosis Clinic" (WO 2021/231374 A1). The author has no other competing interests to declare. Alhaji Cherif: AC is an employee of the Renal Research Institute, a wholly owned subsidiary of Fresenius Medical Care. AC is an inventor on a patent application named "Virtual Osteoporosis Clinic" (WO 2021/231374 A1). The author has no other competing interests to declare. David A Bushinsky: DAB reports a grant from NIH, personal fees, stock and stock options from Tricida; personal fees from and stock in Amgen; personal fees from Relypsa/Vifor/Fresenius; personal fees from Sanifit outside the submitted work. The author has no other competing interests to declare. Ariella Mermelstein: AM is an employee of the Renal Research Institute, a wholly owned subsidiary of Fresenius Medical Care. The author has no other competing interests to declare. Jochen G Raimann: JGR is an employee of the Renal Research Institute, a wholly owned subsidiary of Fresenius Medical Care. The author has no other competing interests to declare. Peter Kotanko: PK is an employee of the Renal Research Institute, a wholly owned subsidiary of Fresenius Medical Care. PK holds stock in Fresenius Medical Care. PK is an inventor on a patent application named "Virtual Osteoporosis Clinic" (WO 2021/231374 A1). The author has no other competing interests to declare.

### Funding

No external funding was received for this work.

### Author contributions

David J Jörg, Conceptualization, Data curation, Formal analysis, Investigation, Methodology, Software, Validation, Visualization, Writing – original draft, Writing – review and editing; Doris H Fuertinger, Conceptualization, Writing – review and editing; Alhaji Cherif, Methodology, Writing – review and editing; David A Bushinsky, Biological and clinical expertise, Writing – review and editing; Ariella Mermelstein, Jochen G Raimann, Data curation, Writing – review and editing; Peter Kotanko, Biological and clinical expertise, Conceptualization, Investigation, Methodology, Visualization, Writing – review and editing

### Author ORCIDs

David J Jörg http://orcid.org/0000-0001-5960-0260
Doris H Fuertinger http://orcid.org/0000-0001-6321-2367

Peter Kotanko  http://orcid.org/0000-0003-4373-412X

**Decision letter and Author response**
Decision letter https://doi.org/10.7554/eLife.76228.sa1
Author response https://doi.org/10.7554/eLife.76228.sa2

## Additional files

### Supplementary files
- Transparent reporting form
- Source code 1. Simulation codes needed to reproduce the results in the paper.

### Data availability
The current manuscript is a computational study, so no data have been generated for this manuscript. Modelling code including the code needed to produce simulation-related figures is part of the Source Code Files. Digitised data from previously published scientific articles are also part of the Source Code Files. All original source publications are specified in Appendix 3 Table 2 of the manuscript. Data were preprocessed as described in the 'Methods' section of the manuscript. Model parameters are also part of the Source Code Files. In addition, they are provided in Appendix 3 Table 4 of the manuscript.

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

## Appendix 1

## Model of long-term bone remodeling and osteoporosis

The model of bone remodeling underlying the present description of osteoporosis and its treatment was built with the aim to provide a minimal set of dynamic components necessary to quantitatively capture population averages of both aging-related changes in bone turnover and the response to osteoporosis medications with different mechanisms of action. The model is partitioned into a core model describing the patient physiology (*Appendix 3—table 1*) and separate extensions for different drug classes (*Appendix 3—table 2*).

The model is compartment-based and describes average cell densities, bone densities, and average concentrations of signaling factors within a 'representative bone remodeling unit (BRU)', that is, a fictitious BRU corresponding to an average over the considered bone type. All model variables are treated as nondimensional quantities, that is, the model only addresses relative changes in all variables, which simplifies the model structure and reduces the number of free parameters.

### Model description

The model describes the dynamics of the cell densities of pre-osteoclasts ($\rho_{C*}$), pre-osteoblasts ($\rho_{B*}$), osteoclasts ($\rho_C$), osteoblasts ($\rho_B$), osteocytes ($\rho_Y$), as well as functional sclerostin levels ($s$), total bone density ($\rho_b$), and BMC ($c_b$). Functional estrogen levels ($e$) are provided as an explicitly age-dependent function, described further below. Moreover, the model includes a 'resorption signal' ($r$) corresponding to the composite concentration of bone matrix-derived signaling factors (TGFβ, BMPs, PDGF, IGFs, and FGFs) released upon bone resorption. Sclerostin, estrogen, and the resorption signal act as regulatory factors that modulate the rates of cell proliferation, differentiation, and apoptosis, as well as their bone-forming and -resorbing activity; for notational convenience, they are summarized in the vector $\phi = (s, e, r)$. Rates that depend on $\phi$ are denoted with a tilde.

The dynamics of the cell density variables is given by

$$\dot\rho_{C*} = \tilde\mu_{C*}(\phi) - f_{C*\to C}(\phi) - \tilde\eta_{C*}(\phi)\rho_{C*} , \tag{1}$$

$$\dot\rho_{B*} = \tilde\mu_{B*}(\phi) - f_{B*\to B}(\phi) - \tilde\eta_{B*}(\phi)\rho_{B*} , \tag{2}$$

$$\dot\rho_C = f_{C*\to C}(\phi) - \tilde\eta_C(\phi)\rho_C , \tag{3}$$

$$\dot\rho_B = f_{B*\to B}(\phi) - \tilde\eta_B(\phi)\rho_B - f_{B\to Y}(\phi) , \tag{4}$$

$$\dot\rho_Y = f_{B\to Y}(\phi) - \tilde\eta_Y(\phi)\rho_Y, \tag{5}$$

where dots denote time derivatives, $\tilde\mu_x$ denotes the formation rate of cell population $x$, and $\tilde\eta_x$ denotes its apoptosis rate. Differentiation or conversion from one cell type to another is described by the absolute differentiation rates $f_{x\to y}$ of cell population $x$ giving rise to cells of type $y$; they have the generic form $f_{x\to y}(\phi) = \tilde\omega_x(\phi)\rho_x$, with $\tilde\omega_x$ denoting the differentiation rate. All rates are functions of the regulatory factors $\phi$ as indicated.

Sclerostin is synthesized by osteocytes; the dynamics of Sclerostin levels is given by

$$\dot s = \tilde\alpha_s(\phi)\rho_Y - \tilde\kappa_s(\phi)s , \tag{6}$$

where $\tilde\alpha_s$ and $\tilde\kappa_s$ denote the synthesis and degradation rate, respectively.

The dynamics of the total bone density (BD) follows:

$$\begin{aligned} \dot\rho_b &= b^+ - b^- , \\ b^+ &= \tilde\lambda_B(\phi)\rho_B , \\ b^- &= \tilde\lambda_C(\phi)\rho_C , \end{aligned} \tag{7}$$

where $b^+$ and $b^-$ are the absolute rates of bone formation and resorption, respectively, with $\tilde\lambda_B$ and $\tilde\lambda_C$ being the formation and resorption rates per unit osteoblast and osteoclast density, respectively.

The BMC follows the dynamics

$$\dot c_b = \tilde\gamma(\phi)[\tilde c_0(\phi) - c_b] , \tag{8}$$

where $\tilde{c}_0$ is the steady-state homeostatic BMC, and $\tilde{\gamma}$ is the equilibration rate.

Based on the physiological foundations summarized in the Section 'Mechanisms of bone turnover and its regulation' in the main text, we now specify the functional form of all rates. Upregulation and downregulation through a regulatory factor with concentration $x$ are described by a multiplicative or additive contribution $g^+(x/x_0)$ or $g^-(x/x_0)$, respectively, where $x_0$ is the threshold concentration at which half-effect is reached (EC$_{50}$) and where $g^{\pm}$ are monotonic, saturating functions of the Hill-type (**Keener and Sneyd, 2009**),

$$g^+(u) = \frac{u}{1+u} , \qquad g^-(u) = \frac{1}{1+u} ,$$

Using these conventions, the dependencies of rates on regulatory factors are given by

$$\tilde{\mu}_{C*}(\phi) = 1 , \qquad\qquad \tilde{\mu}_{B*}(\phi) = 1 , \tag{9}$$

$$\tilde{\omega}_{C*}(\phi) = g^-\left(\frac{e}{e_{C*}}\right)g^+\left(\frac{s}{s_{C*}}\right)\omega_{C*} , \qquad \tilde{\omega}_{B*}(\phi) = g^-\left(\frac{s}{s_{B*}}\right)\omega_{B*} , \tag{10}$$

$$\tilde{\omega}_B(\phi) = \omega_B , \tag{11}$$

$$\tilde{\eta}_{C*}(\phi) = 0 , \qquad\qquad \tilde{\eta}_{B*}(\phi) = 0 , \tag{12}$$

$$\tilde{\eta}_C(\phi) = \left[1 + \nu_C g^+\left(\frac{e}{e_C}\right)g^+\left(\frac{r}{r_C}\right)\right]\eta_C , \qquad \tilde{\eta}_B(\phi) = \eta_B , \tag{13}$$

$$\tilde{\eta}_Y(\phi) = \eta_Y , \tag{14}$$

$$\tilde{\alpha}_s(\phi) = g^-\left(\frac{e}{e_s}\right) , \qquad\qquad \tilde{\kappa}_s(\phi) = \kappa_s , \tag{15}$$

$$\tilde{\lambda}_B(\phi) = \lambda_B g^-\left(\frac{s}{s_\Omega}\right)\left[1 + \nu_\Omega g^+\left(\frac{r}{r_\Omega}\right)\right] , \qquad \tilde{\lambda}_C(\phi) = \lambda_C , \tag{16}$$

$$\tilde{\gamma}(\phi) = \gamma , \qquad\qquad \tilde{c}_0(\phi) = c_0 , \tag{17}$$

where rates without a tilde denote model parameters. A full list of parameters introduced here and their description is provided in **Appendix 3—table 4**. This regulatory scheme comprises a multitude of interactions, many of which are simplified effective representations of indirect molecular mechanisms (e.g., through the RANK–RANKL–OPG pathway as described below). In the model building process, the effects of additional regulatory elements (not presented here) were probed and found to be nonessential within the scope of the present modeling aim or nonidentifiable with regard to related parameter values. Specific choices for regulatory interactions were partly based on insights from animal and culture studies and are motivated as follows:

- *Equation 9*: Pre-osteoclasts and pre-osteoblasts are produced at constant rates, a simplifying assumption reflecting the fact that the main function of these populations in the present context is to provide a dynamic reservoir for the rapid supply with active osteoclasts and osteoblasts, respectively.
- *Equation 10*: Estrogen suppresses pre-osteoclast to osteoclast differentiation; a consequence of suppression of RANKL expression (**Streicher et al., 2017**). Sclerostin upregulates pre-osteoclast to osteoclast differentiation; a consequence of upregulation of RANKL expression and downregulation of OPG expression (**Wijenayaka et al., 2011**). Sclerostin downregulates osteoblastogenesis; a consequence of the inhibition of osteoblast differentiation mediated by bone morphogenetic protein 2 (BMP2) and Wnt3a and possibly other pathways (**Winkler et al., 2005**; **Thouverey and Caverzasio, 2015**).
- *Equation 11*: Sclerostin downregulates osteoblast to osteocyte conversion (**Atkins et al., 2011**).
- *Equation 13*: Estrogen and the resorption signal induce osteoclast apoptosis; estrogen has been reported to induce osteoclast apoptosis both directly and mediated by TGFβ (**Kameda et al., 1997**; **Hughes et al., 1996**); TGFβ has been shown to upregulate Bim, a member of the (pro-apoptotic) Bcl2 family (**Houde et al., 2009**).
- *Equation 15*: Estrogen reduces sclerostin production (**Mödder et al., 2011**).

- *Equation 16*: Sclerostin downregulates bone formation (*Li et al., 2008*; *Atkins et al., 2011*). The resorption signal upregulates bone formation; a consequence of TGFβ1 enhancing bone collagen synthesis (*Rydziel et al., 1997b*); furthermore, TGFβ1, skeletal BMPs, and IGF1 have been reported to inhibit collagenase-3 expression in osteoblasts (*Rydziel et al., 1997b*; *Gazzerro et al., 1999*; *Rydziel et al., 1997a*).

Estrogen concentration is described through its explicit age dependence. Clinical data reported by *Sowers et al., 2008* were used to construct a function capturing the key features of age-dependent decline of serum estradiol levels:

$$e(t) = \begin{cases} 1 & t < t_\mathrm{e} \\ \dfrac{1}{1 + (t - t_\mathrm{e})/\tau_\mathrm{e}} & t \geq t_\mathrm{e} \end{cases} . \tag{18}$$

where $t$ denotes time. The parameters $t_\mathrm{e}$ and $\tau_\mathrm{e}$ denote the age at the onset of estradiol decline and a characteristic time scale of the decline, respectively. The time scale $\tau_\mathrm{e}$ was determined using a fit of the function to the data reported in *Sowers et al., 2008* (*Appendix 1—figure 1A*, *Appendix 3—table 4*).

The resorption signal $r$ corresponds to the concentration of bone matrix-derived signaling factors released upon bone resorption. Assuming a release rate proportional to the bone resorption rate $b^-$ and first-order degradation, we consider a highly simplified dynamics of the type $\dot{r} = b^- - \kappa r$, where $\kappa$ is an effective average degradation rate of the components of the resorption signal. Given that the time scale of degradation, $\kappa^{-1}$, is much shorter (minutes to hours) than the time scale of osteoclast formation and death (weeks), the instantaneous concentration can be approximated to always follow its steady state, $r \approx b^-/\kappa$, which is proportional to the osteoclast density, $r \propto b^- \propto \rho^\mathrm{C}$, via *Equation 7*. Since the resorption signal acts as a regulator of bone formation and is rescaled by individual concentration thresholds (see *Equation 9–Equation 17*), the proportionality constant can be absorbed in these thresholds, which enables us to set $r = \rho^\mathrm{C}$. Thus, the resorption signal concentration is approximated by the osteoclast density, so that no additional dynamic variable is required.

## Description of BMD, bone turnover rate, and BTMs

To compare the model output to clinical data, we relate model variables to clinical observables frequently measured in clinical trials such as BMD and established biomarkers of bone turnover. The BMD follows from the model state as the product of total bone density and BMC:

$$\mathrm{BMD} = \rho_\mathrm{b} c_\mathrm{b} . \tag{19}$$

In our model, levels of BTMs such as the bone formation markers P1NP and BSAP and the bone resorption marker CTX are related to the rates $b^+$ and $b^-$ of bone formation and resorption, respectively (see *Equation 7*). Here, we relate the BTMs P1NP, BSAP, and CTX to bone turnover rates by power laws with marker-specific exponents:

$$\begin{aligned} \theta_\mathrm{BSAP} &= (b^+)^{q_\mathrm{BSAP}} , \\ \theta_\mathrm{P1NP} &= (b^+)^{q_\mathrm{P1NP}} , \\ \theta_\mathrm{CTX} &= (b^-)^{q_\mathrm{CTX}} . \end{aligned} \tag{20}$$

The exponents $q_\mathrm{x}$ are obtained as fit parameters using clinical trial data, as described further below.

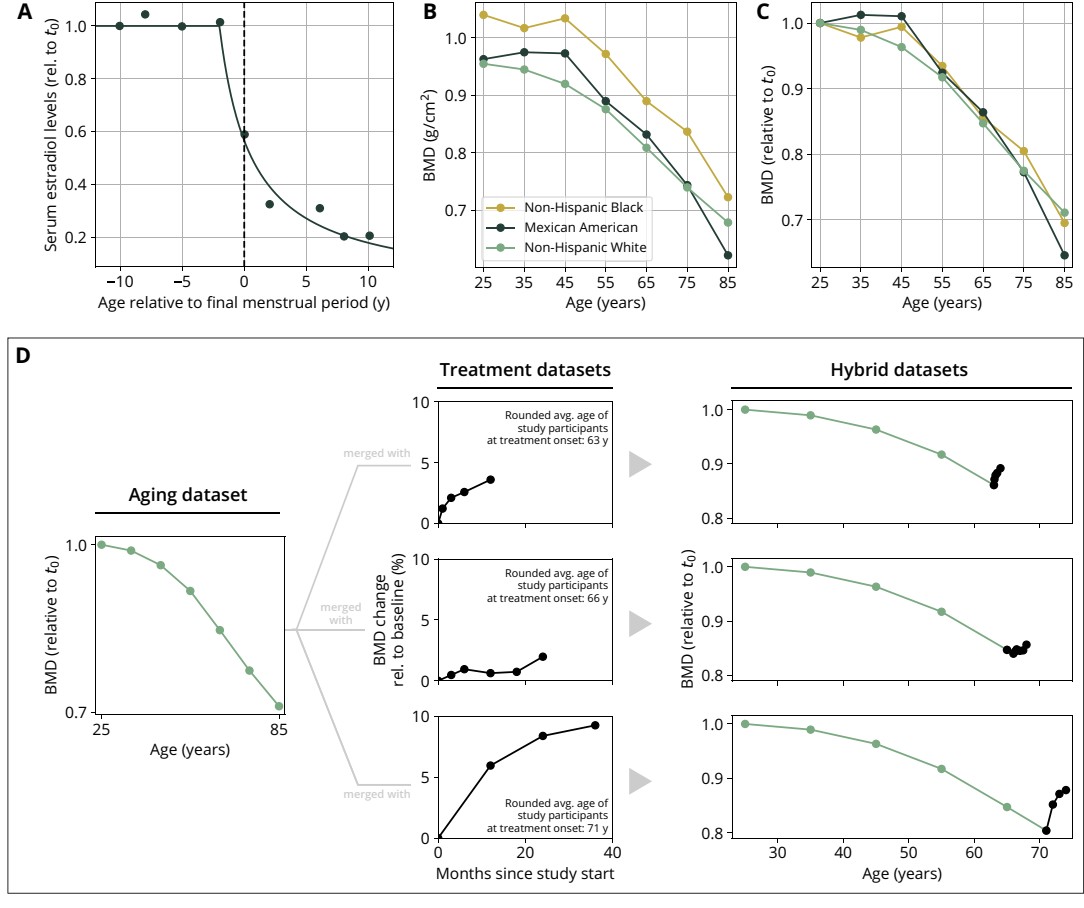

**Appendix 1—figure 1.** Parameterization of the aging behavior and creation of hybrid aging/treatment datasets for model calibration and validation. (**A**) Age dependence of estradiol serum levels. Clinical data (dots) modified from *Sowers et al., 2008*. The curve shows a fit of the function given by *Equation 18* to determine the parameter $\tau_e$ (*Appendix 3—table 4*). (**B**) Bone mineral density (BMD) age dependence for different ethnic groups as indicated. Data modified from *Looker et al., 1998*; reported age bin averages have been used to represent the center of the age bin. (**C**) BMD age dependence shown in panel (**B**), where all curves have been normalized to their earliest value ($t_0 = 25y$). (**D**) Schematic of how hybrid aging/treatment datasets were generated by merging the same aging dataset with different treatment datasets; for details, see 'Methods.'

## Appendix 2

## Model extensions for medications

We include a dynamic description of several drug classes through separate model extensions, which depend on the functional drug concentration. The pharmacodynamic description is drug-specific and represents the individual mechanism of action of the respective drug class. For the pharmacokinetic description of each drug, we resort to simple first-order kinetics with drug-specific half-lives, which reduces the amount of model parameters. More detailed pharmacokinetic descriptions involve drug absorption and transfer between different body compartments, depending on the route of administration (oral, intravenous, or subcutaneous). However, simulations of the calibrated model demonstrate that first-order kinetics yields an effective approximation of the pharmacokinetic features essential to capture a drug's long-term effects on bone remodeling, as suggested by comparisons of simulated and measured BTM concentrations (*Figure 2*, *Appendix 3—figure 1*). A patient's systemic concentration of a medication is represented by an effective (dimensionless) variable $\psi$ that indicates the relative concentration of the medication. Typically, $\psi$ is given in multiples of a threshold that parameterizes the effect of the drug (such as $EC_{50}$)—the precise interpretation of $\psi$ depends on the model extension that describes the pharmacodynamics of the drug; see 'Pharmacodynamics for specific medications'.

### Pharmacokinetics

The pharmacokinetics of a drug $x$ that is administered in intervals of weeks or months is described by two parameters: the efficacy $E_x$ and the half-life $T_x$. Given repeated administrations with doses $c_1, \ldots, c_n$ at times $t_1, \ldots, t_n$, the efficacy-weighted concentration variable of the drug $x$ therefore follows the exponential kinetics

$$\psi_x(t) = E_x \sum_{i=1}^{n} c_i 2^{-(t-t_i)/T_x} \Theta(t - t_i) \,, \tag{21}$$

where $\Theta$ is the Heaviside function, defined by

$$\Theta(t) = \begin{cases} 0 & t < 0 \\ 1 & t \geq 0 \end{cases} \,.$$

Drugs that are administered more frequently (e.g., daily or weekly) are more efficiently captured in a quasi-continuous scheme. The dynamics of BMD and BTM levels is much more inert than such fast administration/degradation dynamics and is well-described by their effective average action. In this quasi-continuous scheme, the drug is considered to be administered at a given average rate for a specified amount of time, so that its concentration evolves according to the dynamic equation

$$\frac{d\psi_x}{dt} = E_x \sum_{i=1}^{n} c_i \Theta(t - t_i) \Theta(t_i^* - t) - \frac{\ln 2}{T_x} \psi_x, \tag{22}$$

with initial condition $\psi_x(t)|_{t \to -\infty} = 0$, where $c_i$ are doses per unit time, and where $t_i$ and $t_i^*$ are the start and end times of a treatment period, respectively. (Numerically, the quasi-continuous scheme has the advantage that model simulations do not have to resort to extremely small integration time steps to capture the details of short-term drug degradation, which considerably improves runtime.)

Different drugs $x_1, x_2, \ldots$ of the same class (sclerostin inhibitors, bisphosphonates, PTH analogs, etc.) are considered through an effective concentration equivalent that is the sum of the efficacy-weighted doses of different drugs:

$$\Psi(t) = \sum_i \psi_{x_i}(t) \,, \tag{23}$$

where the $\psi_{x_i}$ are given by *Equation 21* in the case of discrete doses and *Equation 22* in the case of quasi-continuous dosing.

## Pharmacodynamics for specific medications

We now introduce separate model extensions that embody the essential mechanisms of action of different drug classes.

### RANKL antibodies

Denosumab is a monoclonal antibody (mAb) that binds with affinity to receptor activator of NF-$\kappa$B ligand (RANKL) and blocks its interactions with RANK (*Kostenuik et al., 2009*), which, in turn, decreases osteoclast formation (*Boyce and Xing, 2008*). Moreover, denosumab has been suggested to enable increased bone tissue mineralization, which leads not only to a halt of the BMD decline but also to a long-term increase in BMD (*Scheiner et al., 2014*). We include these mechanisms in our model through a modification of the reference pre-osteoclast differentiation rate $\omega_{C*}$ to include a downregulation by denosumab and the steady-state BMC $c_0$ to include an upregulation through denosumab in *Equation 10* and *Equation 17*, respectively:

$$
\begin{aligned}
\omega_{C*} &\to [1 - \beta_{C*}^{rAb} g^+(\Psi_{rAb})]\omega_{C*} \,, \\
c_0 &\to c_0 + \beta_{b}^{rAb} g^+(\Psi_{rAb}) \,,
\end{aligned}
\tag{24}
$$

where $\Psi_{rAb}$ is the RANKL antibody concentration in multiples of the half-maximal effective concentration ($EC_{50}$), determined through *Equation 21* and *Equation 23*. For simplicity, $EC_{50}$ for the regulation of both differentiation and mineralization are taken to be identical, which is justified a posteriori by showing its effectivity in approximating the drug action. The scaling factors $\beta_{C*}^{rAb}$ and $\beta_{b}^{rAb}$ parameterize the respective maximum effect strength and are subject to the constraints $\beta_{C*}^{rAb} < 1$ and $c_0 + \beta_{b}^{rAb} < 1$ to ensure positive rates and BMCs between 0 and 100%.

### Sclerostin antibodies

Romosozumab and blosozumab are mAbs that bind to sclerostin and prevent its inhibitory effects on bone formation (*Recknor et al., 2015*; *Lim and Bolster, 2017*; *McClung et al., 2018*). (Note that blosozumab was not approved for osteoporosis treatment at the time this manuscript was written.) Accordingly, we represent the mechanism of action of sclerostin antibodies by adding a new variable $s^*$ corresponding to the level of antibody-bound sclerostin and adding the dynamics of antibody-binding and unbinding in *Equation 6*,

$$
\begin{aligned}
\dot{s} &\to \dot{s} - \kappa_s \Psi_{sAb} s + \delta_s s^* \,, \\
\dot{s}^* &= \kappa_s \Psi_{sAb} s - (\delta_s + \kappa_s) s^* \,,
\end{aligned}
\tag{25}
$$

where $\Psi_{sAb}$ is the effective sclerostin antibody concentration equivalent determined through *Equation 21* and *Equation 23*. Here, $\kappa_s$ denotes the sclerostin degradation rate and $\delta_s$ is the sclerostin/antibody binding rate; this parameterization implies that $\Psi_{sAb}$ is given in multiples of the effective antibody levels needed to achieve a binding rate equal to the unperturbed degradation grade of sclerostin.

### Bisphosphonates

Bisphosphonates (like alendronate, ibandronate, risedronate, and zoledronate) bind to hydroxyapatite on the bone surface, thereby preventing osteoclasts from bone resorption; they further inhibit osteoclast-mediated bone resorption by promoting osteoclast apoptosis (*Sato et al., 1991*; *Rodan and Fleisch, 1996*). To simplify the model extension, we effectively represent the mechanism of action of bisphosphonates through the upregulation of the osteoclast apoptosis rate in *Equation 13*:

$$
\eta_C \to \eta_C + g^+(\Psi_{bp})\eta_C^{bp} \,,
\tag{26}
$$

where $\Psi_{bp}$ is the effective bisphosphonate concentration equivalent determined through *Equation 22* and *Equation 23*, and $\eta_C^{bp}$ is the maximum additional apoptosis rate caused by the presence of bisphosphonates.

## PTH analogs

Teriparatide and abaloparatide are recombinant human PTH analogs (*Jiang et al., 2003*; *Hattersley et al., 2016*). PTH is known to have an anabolic effect on bone if administered intermittently while exerting a catabolic effect if administered continuously (*Tam et al., 1982*; *Hock and Gera, 1992*). Here, we consider an effective representation of the action of PTH in the anabolic regime only; this leads to a highly simplified and efficient description of the effective action of PTH therapies on bone turnover. However, it implies that the scope of our model is restricted to anabolic administration schemes and cannot be expected to yield correct results if probed in inappropriate regimes. In the anabolic regime, teriparatide downregulates osteoblast apoptosis (*Jilka et al., 1999*). Moreover, bone turnover markers show a marked increase early after treatment start but decline while drug administration remains unaltered (*Leder et al., 2014*, see *Appendix 3—figure 1*); such an effect is achieved in our model by rapid upregulation of osteoclast/osteoblast differentiation. We therefore also include a regulatory effect on osteoclast differentiation (which indirectly affects osteoblast differentiation as well):

$$
\begin{aligned}
\eta_{\mathrm{B}} &\rightarrow [1 - \beta_{\mathrm{B}}^{\mathrm{pth}} g^{+}(\Psi_{\mathrm{pth}})]\eta_{\mathrm{B}}, \\
\omega_{\mathrm{C}^{*}} &\rightarrow [1 + \beta_{\mathrm{C}^{*}}^{\mathrm{pth}} g^{+}(\Psi_{\mathrm{pth}})]\omega_{\mathrm{C}^{*}},
\end{aligned}
\tag{27}
$$

where $\Psi_{\mathrm{pth}}$ is the effective teriparatide concentration equivalent determined through *Equation 22* and *Equation 23*, and the parameters $\beta_{\mathrm{B}}^{\mathrm{pth}}$ and $\beta_{\mathrm{C}^{*}}^{\mathrm{pth}}$ parameterize the maximum effect strength of osteoblast apoptosis and pre-osteoclast differentiation, respectively.

# Appendix 3

## Simulations and parameter fits

### Simulation protocol for aging and treatment

A model simulation was implemented in Python using standard NumPy and SciPy packages (*Oliphant, 2006*; *Virtanen et al., 2020*) and solved using SciPy's 'solve_ivp' function with BDF solver. To compare how the model predicts the bone turnover dynamics of a hybrid aging/treatment dataset (see 'Methods' and *Appendix 1—figure 1D*) for a given set of model parameters, simulations were structured as follows. Drug dosing information of the corresponding dataset was provided to the model through the set of administered doses and the administration times: For the case of discrete dosing, dosing information consisted of doses $c_i$ and administration times $t_i$ entering *Equation 21* (used for the drugs blosozumab, romosozumab, and denosumab). For the case of quasi-continuous dosing, dosing information consisted of doses per unit time $c_i$ and time windows $[t_i, t_i^*]$ entering *Equation 22* (used for the drugs alendronate and teriparatide).

The model was initialized at $t = 0$ in its steady state for all dynamic variables (i.e., the state for which all time derivatives are zero); except for the bone density $\rho_b$, which does not possess a unique steady state and which was set to unity to represent peak bone density. We then simulated the model until well after the treatment period. All aging-related effects in the model were mediated by explicitly time-dependent auxiliary functions, as explained in Appendix 1 . To compare simulation results with clinical data, all relevant model variables were rescaled such that the first recorded data point in the corresponding clinical dataset coincided with the corresponding time point in the simulation, so that relative changes from a reference time point could be compared.

### Parameter fits

To systematically fit model parameters, we defined a cost function that takes into account multiple fit quantities depending on their availability in the datasets. For a hybrid dataset $\alpha$ and clinical quantity $\beta$ (BMD and serum levels of CTX, P1NP, and BSAP), we defined the distance function

$$D^{\alpha\beta} = \frac{1}{\sum_i w_i^{\alpha\beta}} \sum_i w_i^{\alpha\beta} z_i^{\alpha\beta} \left( x_i^{\alpha\beta} - \hat{x}_i^{\alpha\beta} \right)^2 , \tag{28}$$

where $x_i$ denotes the clinical data point at time point  , $\hat{x}_i$ denotes the respective simulated data point, $w_i$ denotes the relative weight of the respective data point depending on its certainty, and $z_i$ denotes the relative weight depending on the time interval represented by the respective data point. For BTMs, we used the weights $w_i^{\alpha\beta} = 1/(1 + e_i^{\alpha\beta})$, where $e_i^{\alpha\beta}$ is the mean of the upper and lower error bars of the respective quantity $\beta$; for the BMD, we used unit weights ($w_i^{\alpha,\text{BMD}} = 1$). To account for the fact that time intervals between data points may vastly differ (e.g., between the coarsely sampled aging dataset and the densely sampled treatment datasets), we included an interval-dependent weighting factor $z_i$, such that each data point was weighted by the average distance to its neighboring data points: The time interval-related weighting factors $z_i^{\alpha\beta}$ were defined as $z_i = (\delta_{i-1} + \delta_i)/2$ $(1 \leq i \leq n)$, where $\delta_i = t_{i+1} - t_i$ $(1 \leq i \leq n-1)$, $\delta_0 = \delta_1$, $\delta_n = \delta_{n-1}$ and $t_i$ denotes the time point of measurement  .

We defined the combined cost function over all considered datasets as

$$J = \sum_{\alpha\beta} W^\beta D^{\alpha\beta} , \tag{29}$$

where $W^\beta$ is an additional weighting factor that determines the relative importance of the different fit quantities in the cost function (*Appendix 3—table 1*).

In a first step, all fit parameters were manually adjusted for the model to exhibit a roughly sensible aging behavior. In a second step, a selected subset of hybrid aging/treatment datasets (see 'Methods' and *Appendix 1—figure 1D*) were used to fit all free model parameters. To perform the fits, we used a Covariance Matrix Adaptation Evolution Strategy via the Python package 'pycma' (*Hansen et al., 2019*). Results of the parameter fits are shown in *Appendix 3—figure 1*. The full list of fit parameters, including their final fit values, is given in *Appendix 3—table 4*.

**Appendix 3—table 1.** Values of fit weights $W^\beta$ used in *Equation 29*.

| Weight | Value |
| --- | --- |
| $W^{(\text{BMD})}$ | 300 |
| $W^{(\text{CTX})}$ | 1 |
| $W^{(\text{P1NP})}$ | 1 |
| $W^{(\text{BSAP})}$ | 1 |

BMD, bone mineral density; CTX, C-terminal telopeptide; P1NP, procollagen type 1 amino-terminal propeptide; BSAP, bone-specific alkaline phosphatase

## Goodness-of-fit measures

To assess the goodness of the parameter fit and model predictions, we considered complementary goodness measures. The MAPE between clinical and simulated results is defined by

$$G_{\text{MAPE}} = \frac{1}{n} \sum_{i=1}^{n} \frac{|x_i - \hat{x}(t_i)|}{x_i} \,, \tag{30}$$

where the sum runs over all clinically recorded time points for the respective quantity (baseline changes of BMD and BTM levels), $x_i$ denotes the clinical data point, $t_i$ the time it was taken, and $\hat{x}(t)$ denotes the model result, which is a continuous function of time.

As a complementary measure, we introduce a 'windowed minimal absolute percentage error' (WMAPE), which indicates the mean minimal distance between model results and the data within a time window around the data point that reflects the average time spacing between data points. Formally, the WMAPE is given by

$$G_{\text{WMAPE}} = \frac{1}{n} \sum_{i=1}^{n} \frac{1}{x_i} \min_{t \in [t_i - \tau, t_i + \tau]} |x_i - \hat{x}(t)| \,, \tag{31}$$

Here, we choose the time window $\tau$ as half the median distance between data points, $\tau = \text{median}_i(t_i - t_{i-1})/2$.

**Appendix 3—table 2.** Data sources used to calibrate and validate the model.

Columns titled 'Figure(s)' indicate the plot panels in the respective publication that were digitized. BMD always refers to total hip bone mineral density.

| | | | Figure(s) | | | | Table(s) |
| --- | --- | --- | --- | --- | --- | --- | --- |
| Publication | Medication(s) | Dosings | BMD | CTX | P1NP | BSAP | BMD |
| *Black et al., 2006* | Alendronate | 5–10 mg Q1D | 2 | 3 | 3 | — | — |
| *Bone et al., 2011* | Denosumab | 60 mg Q6M | 3b | 4b | 4a | — | — |
| *Cosman et al., 2016* | Romosozumab | 210 mg Q1M | 3b | 3e | 3d | — | — |
| | Denosumab | 60 mg Q6M | 3b | 3e | 3d | — | — |
| *Leder et al., 2014* | Teriparatide | 20 mcg Q1D | 2d | 4c,f | 4b,e | — | — |
| *Leder et al., 2015* | Teriparatide | 20 mcg Q1D | 3 | 4 | — | — | — |
| | Denosumab | 60 mg Q6M | 3 | 4 | — | — | — |
| *Lewiecki et al., 2019* | Romosozumab | 210 mg Q1M | 3b | — | — | — | — |
| | Denosumab | 60 mg Q6M | 3b | — | — | — | — |
| *Looker et al., 1998* | [Age-dependent BMD] | — | — | — | — | — | 7 |
| *McClung et al., 2006* | Denosumab | 6 mg Q3M, 14 mg Q6M, 210 mg Q6M | 2b | 2e | — | 2f | — |
| *McClung et al., 2017* | Denosumab | 6–14 mg Q3M, 14–210 mg Q6M | 2b | — | — | — | — |

*Appendix 3—table 2 Continued on next page*

*Appendix 3—table 2 Continued*

| Publication | Medication(s) | Dosings | Figure(s) BMD | CTX | P1NP | BSAP | Table(s) BMD |
|---|---|---|---|---|---|---|---|
| *McClung et al., 2018* | Romosozumab | 140 mg Q1M, 210 mg Q1M | 3c | 4b | 4a | — | — |
| | Denosumab | 60 mg Q6M | 3c,d | 4b,d | 4a,c | — | — |
| | Alendronate | 70 mg Q1W | 3d | 4d | 4c | — | — |
| *Recknor et al., 2015* | Blosozumab | 180 mg Q4W, 180 mg Q2W, 270 mg Q2W | 3b | 4d | 4a | — | — |
| *Saag et al., 2017* | Alendronate | 70 mg Q1W | 3b | 3d | 3c | — | — |

Q, every; M, month; D, day; W, week; CTX, C-terminal telopeptide; P1NP, procollagen type 1 amino-terminal propeptide; BSAP, bone-specific alkaline phosphatase.

**Appendix 3—table 3.** Goodness-of-fit measures for calibration and validation datasets. Mean absolute percentage error (MAPE) and windowed minimal absolute percentage error (WMAPE) as defined in *Equation 30* and *Equation 31*, respectively. The column 'Shown in' indicates the figure in this article that shows the respective simulation and data plot.

| Medication(s) | Data ref. | MAPE BMD (%) | CTX (%) | P1NP (%) | BSAP (%) | WMAPE BMD (%) | CTX (%) | P1NP (%) | BSAP (%) | Shown in |
|---|---|---|---|---|---|---|---|---|---|---|
| **Calibration datasets** | | | | | | | | | | |
| Alendronate 5–10 mg Q1D | *Black et al., 2006* | 0.9 | 7.0 | 21.1 | — | 0.6 | 4.7 | 13.5 | — | *Appendix 3—figure 1* |
| Alendronate 70 mg Q1W | *Saag et al., 2017* | 0.5 | 34.4 | 13.7 | — | 0.2 | 20.3 | 10.2 | — | *Appendix 3—figure 1* |
| Blosozumab 180 mg Q4W | *Recknor et al., 2015* | 0.7 | 16.5 | 23.8 | — | 0.4 | 10.8 | 18.7 | — | *Appendix 3—figure 1* |
| Blosozumab 270 mg Q2W | *Recknor et al., 2015* | 0.3 | 26.8 | 13.2 | — | 0.2 | 15.8 | 4.8 | — | *Appendix 3—figure 1* |
| Denosumab 14 mg Q3M → denosumab 60 mg Q6M | *McClung et al., 2017* | 0.3 | — | — | — | 0.1 | — | — | — | *Appendix 3—figure 1* |
| Denosumab 14 mg Q6M | *McClung et al., 2006* | 0.2 | 73.9 | — | 17.0 | 0.0 | 46.3 | — | 0.2 | *Appendix 3—figure 1* |
| Placebo | *Recknor et al., 2015* | 0.4 | 7.3 | 15.4 | — | 0.3 | 7.2 | 15.3 | — | *Appendix 3—figure 1* |
| Teriparatide 20 mcg Q1D | *Leder et al., 2014* | 0.4 | 17.1 | 8.5 | — | 0.2 | 9.1 | 1.6 | — | *Appendix 3—figure 1* |
| Teriparatide 20 mcg Q1D→ denosumab 60 mg Q6M | *Leder et al., 2015* | 0.4 | 65.5 | — | — | 0.2 | 8.6 | — | — | *Appendix 3—figure 1* |
| **Validation datasets** | | | | | | | | | | |
| Alendronate 70 mg Q1W → romosozumab 140 mg Q1M → denosumab 60 mg Q6M | *McClung et al., 2018* | 0.5 | 25.4 | 20.0 | — | 0.3 | 17.1 | 3.6 | — | *Figure 2* |
| Blosozumab 180 mg Q2W | *Recknor et al., 2015* | 0.3 | 21.3 | 20.3 | — | 0.1 | 13.2 | 12.6 | — | *Figure 2* |
| Denosumab 60 mg Q6M → teriparatide 20 mcg Q1D | *Leder et al., 2015* | 0.4 | 103.1 | — | — | 0.3 | 44.6 | — | — | *Figure 2* |
| Placebo | *McClung et al., 2018* | 0.6 | 6.2 | 10.5 | — | 0.5 | 6.2 | 10.5 | — | *Figure 2* |
| Placebo→ denosumab 60 mg Q6M | *McClung et al., 2018* | 0.6 | 13.2 | 10.1 | — | 0.4 | 5.1 | 7.4 | — | *Figure 2* |
| Teriparatide 20 mcg Q1D + denosumab 60 mg Q6M → denosumab 60 mg Q6M | *Leder et al., 2015* | 0.7 | 183.7 | — | — | 0.4 | 66.4 | — | — | *Figure 2* |
| Alendronate 70 mg Q1W → romosozumab 140 mg Q1M → placebo | *McClung et al., 2018* | 0.5 | 22.9 | 16.8 | — | 0.4 | 15.1 | 4.2 | — | *Figure 2—figure supplement 1* |
| Placebo→ denosumab 60 mg Q6M | *Cosman et al., 2016* | 0.5 | 81.9 | 9.6 | — | 0.0 | 44.4 | 4.0 | — | *Figure 2—figure supplement 1* |
| Placebo→ denosumab 60 mg Q6M | *Lewiecki et al., 2019* | 0.4 | — | — | — | 0.1 | — | — | — | *Figure 2—figure supplement 1* |
| Placebo→ denosumab 60 mg Q6M | *McClung et al., 2017* | 0.6 | — | — | — | 0.3 | — | — | — | *Figure 2—figure supplement 1* |

*Appendix 3—table 3 Continued on next page*

*Appendix 3—table 3 Continued*

**Validation datasets**

| | | | | | | | | | | |
|---|---|---|---|---|---|---|---|---|---|---|
| Romosozumab 210 mg Q1M → alendronate 70 mg Q1W | *Saag et al., 2017* | 1.0 | 30.0 | 37.1 | — | 0.5 | 19.4 | 11.7 | — | *Figure 2—figure supplement 1* |
| Romosozumab 210 mg Q1M → denosumab 60 mg Q6M | *Cosman et al., 2016* | 1.2 | 92.5 | 22.4 | — | 0.6 | 50.7 | 6.9 | — | *Figure 2—figure supplement 1* |
| Romosozumab 210 mg Q1M → denosumab 60 mg Q6M | *Lewiecki et al., 2019* | 1.0 | — | — | — | 0.5 | — | — | — | *Figure 2—figure supplement 1* |
| Romosozumab 210 mg Q1M → denosumab 60 mg Q6M | *McClung et al., 2018* | 0.6 | 14.2 | 57.8 | — | 0.3 | 6.5 | 15.9 | — | *Figure 2—figure supplement 1* |
| Romosozumab 210 mg Q1M → placebo | *McClung et al., 2018* | 0.7 | 15.0 | 53.3 | — | 0.5 | 10.2 | 18.5 | — | *Figure 2—figure supplement 1* |

Q, every; M, month; D, day; W, week; BMD, bone mineral density; CTX, C-terminal telopeptide; P1NP, procollagen type 1 amino-terminal propeptide; BSAP, bone-specific alkaline phosphatase.

**Appendix 3—table 4.** Full list of parameters of the core model and the medication extensions.

| Parameter | Description | Value | Unit | Origin | Model equation |
|---|---|---|---|---|---|
| **Core model** | | | | | |
| $\omega_{C*}$ | Reference pre-osteoclast to osteoclast differentiation rate | 0.93 | $d^{-1}$ | Calibration | *Equation 10* |
| $e_{C*}$ | Estrogen threshold for downregulation of pre-osteoclast to osteoclast differentiation | 0.94 | 1 | Calibration | *Equation 10* |
| $s_{C*}$ | Sclerostin threshold for upregulation of pre-osteoclast to osteoclast differentiation | $8.60 \times 10^6$ | 1 | Calibration | *Equation 10* |
| $\eta_C$ | Reference osteoclast apoptosis rate | 0.02 | $d^{-1}$ | Calibration | *Equation 13* |
| $e_C$ | Estrogen threshold for upregulation of osteoclast apoptosis | 0.99 | 1 | Calibration | *Equation 13* |
| $r_C$ | Resorption signal threshold for upregulation of osteoclast apoptosis | 10.10 | 1 | Calibration | *Equation 13* |
| $\nu_C$ | Max. rel. effect of regulatory factors on osteoclast apoptosis | $1.23 \times 10^{-4}$ | 1 | Calibration | *Equation 13* |
| $\omega_{B*}$ | Reference pre-osteoblast to osteoblast differentiation rate | 0.32 | $d^{-1}$ | Calibration | *Equation 10* |
| $s_{B*}$ | Sclerostin threshold for downregulation of pre-osteoblast to osteoblast differentiation | $1.63 \times 10^2$ | 1 | Calibration | *Equation 10* |
| $\eta_B$ | Reference osteoblast apoptosis rate | $8.68 \times 10^{-3}$ | $d^{-1}$ | Calibration | *Equation 13* |
| $\omega_B$ | Reference osteoblast to osteocyte conversion rate | $6.24 \times 10^{-4}$ | $d^{-1}$ | Calibration | *Equation 11* |
| $\eta_Y$ | osteocyte apoptosis rate | $1.10 \times 10^{-4}$ | $d^{-1}$ | Estimate | *Equation 14* |
| $\kappa_s$ | Sclerostin degradation rate | 0.05 | $d^{-1}$ | Estimate; see *Suen et al., 2015*; *Ominsky et al., 2015*. | *Equation 15* |
| $e_s$ | Estrogen threshold for downregulation of sclerostin secretion | 9.60 | 1 | Calibration | *Equation 15* |
| $\lambda_C$ | Reference bone resorption rate per unit density osteoclast | $3.82 \times 10^{-6}$ | $d^{-1}$ | Calibration | *Equation 16* |
| $\lambda_B$ | Reference bone formation rate per unit density osteoblast | $1.29 \times 10^{-6}$ | $d^{-1}$ | Calibration | *Equation 16* |
| $s_\Omega$ | Sclerostin threshold for downregulation of bone formation | $3.04 \times 10^3$ | 1 | Calibration | *Equation 16* |

*Appendix 3—table 4 Continued on next page*

*Appendix 3—table 4 Continued*

| Parameter | Description | Value | Unit | Origin | Model equation |
|---|---|---|---|---|---|
| **Core model** | | | | | |
| $r_\Omega$ | Resorption signal threshold for upregulation of bone formation | $1.02 \times 10^3$ | 1 | Calibration | *Equation 16* |
| $\nu_\Omega$ | Max. rel. effect of the resorption signal on bone formation | $1.08 \times 10^2$ | 1 | Calibration | *Equation 16* |
| $\gamma$ | Equilibration rate of the bone mineral content | $6.65 \times 10^{-3}$ | $d^{-1}$ | Calibration | *Equation 17* |
| $c_0$ | Reference bone mineral content | 0.80 | 1 | Estimate | *Equation 17* |
| $t_e$ | Onset of estrogen decline | 50.00 | y | Estimate | *Equation 18* |
| $\tau_e$ | Time scale of estrogen decline | 2.60 | y | Indep. fit (*Appendix 1—figure 1A*) | *Equation 18* |
| **Bone turnover markers** | | | | | |
| $q_{CTX}$ | Exponent relating the bone resorption rate to CTX levels | 1.16 | 1 | Calibration | *Equation 15* |
| $q_{P1NP}$ | Exponent relating the bone formation rate to P1NP levels | 1.45 | 1 | Calibration | *Equation 15* |
| $q_{BSAP}$ | Exponent relating the bone formation rate to BSAP levels | 0.92 | 1 | Calibration | *Equation 15* |
| **Medication extension: sclerostin antibodies** | | | | | |
| $E_{blosozumab}$ | Efficacy: blosozumab | 0.01 | 1 | Calibration | *Equation 21* |
| $T_{blosozumab}$ | Effective half-life: blosozumab | 7.00 | d | $T_{romosozumab}$ | *Equation 21* |
| $E_{romosozumab}$ | Efficacy: romosozumab | 0.01 | 1 | $E_{blosozumab}$ | *Equation 21* |
| $T_{romosozumab}$ | Effective half-life: romosozumab | 7.00 | d | *Solling et al., 2018* | *Equation 21* |
| $\delta_s$ | Sclerostin/antibody unbinding rate | 0.05 | $d^{-1}$ | $\kappa_s$ | *Equation 25* |
| Medication extension: RANKL antibodies | | | | | |
| $E_{denosumab}$ | Efficacy: denosumab | $4.34 \times 10^3$ | 1 | Calibration | *Equation 21* |
| $T_{denosumab}$ | Effective half-life: denosumab | 10.00 | d | *Bekker et al., 2004* | *Equation 21* |
| $\beta_{C*}^{rAb}$ | Max. rel. effect of RANKL antibodies on pre-osteoclast to osteoclast differentiation | 0.87 | 1 | Calibration | *Equation 24* |
| $\beta_b^{rAb}$ | Max. rel. effect of RANKL antibodies on mineralization | 0.02 | 1 | Calibration | *Equation 24* |
| **Medication extension: bisphosphonates** | | | | | |
| $E_{alendronate}$ | Efficacy: alendronate | $2.97 \times 10^{-5}$ | 1 | Calibration | *Equation 22* |
| $T_{alendronate}$ | Effective half-life: alendronate | $1.53 \times 10^2$ | d | Calibration | *Equation 22* |
| $\eta_C^{bp}$ | Max. contribution of bisphosphonates to osteoclast apoptosis rate | 1.00 | $d^{-1}$ | Calibration | *Equation 26* |
| **Medication extension: PTH analogs** | | | | | |
| $E_{teriparatide}$ | Efficacy: teriparatide | 0.27 | 1 | Calibration | *Equation 22* |
| $T_{teriparatide}$ | Effective half-life: teriparatide | 0.04 | d | *Satterwhite et al., 2010* | *Equation 22* |
| $\beta_B^{pth}$ | Max. rel. effect of PTH analogs on osteoblast apoptosis | 1.31 | 1 | Calibration | *Equation 27* |
| $\beta_{C*}^{pth}$ | Max. rel. effect of PTH analogs on pre-osteoclast to osteoclast differentiation | 4.28 | 1 | Calibration | *Equation 27* |

CTX, C-terminal telopeptide; P1NP, procollagen type 1 amino-terminal propeptide; BSAP, bone-specific alkaline phosphatase; PTH, parathyroid hormone.

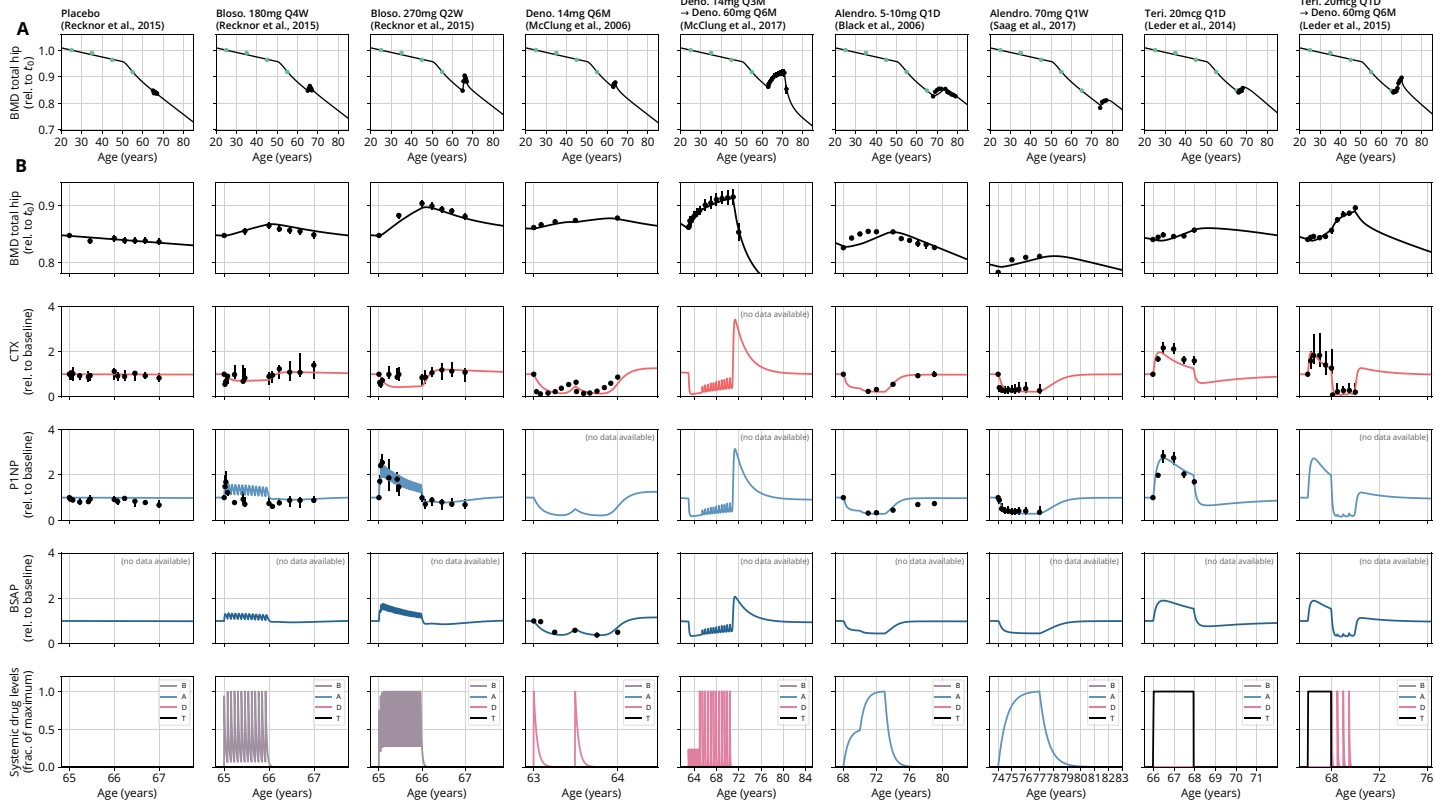

**Appendix 3—figure 1.** Calibration datasets comparing model predictions and clinical data from various studies. All conventions identical to *Figure 2*. Drug administrations are provided in the bottom row. See *Appendix 3—table 2* for a list of data sources and *Appendix 3—table 3* for goodness-of-fit measures. Dosing: mg, milligrams; mcg, micrograms; Q *x* M, dose administered every *x* months; Q *x* W, every *x* weeks; Q *x* D, every *x* days; B, blosozumab; A, alendronate; D, denosumab; T, teriparatide.

