## [Editor Report]

The authors have developed a mathematical framework of drug interventions for postmenopausal osteoporosis using bisphosphonates, parathyroid hormone analogs, romosozumab, and denosumab. After calibrating and validating the model, authors demonstrated a predictive ability for complex clinical scenarios including sequential and parallel drug combinations. These data may be of great help in clinical practice.

---

## [Decision Letter]

**Decision letter after peer review:**

Thank you for submitting your article "Modeling osteoporosis to design and optimize pharmacologic therapies comprising multiple drug types" for consideration by *eLife*. Your article has been reviewed by 1 peer reviewer, and the evaluation has been overseen by a Reviewing Editor and Mone Zaidi as the Senior Editor. The following individual involved in review of your submission has agreed to reveal their identity: Peter Pivonka (Reviewer #1).

Essential revisions:

(1) The discussion needs a lot of revision: it must be clear to authors that the manuscript will be read also by a lot of physicians and a clear, clinical language should be used.

(2) In terms of bone density measurements authors should compare their data with those from Black et al., (JBMR 2018) from the SOF study and Dr Ensrud (JCEM 2022).

The paper is very well written and presents promising results.

The introduction is very comprehensive.

The only additional reference I would suggest is:

Lavaill et al., 2021 "Effects of PTH treatment in osteoporosis – insights from a mechanistic PK-PD model, BMMB, pp1-16, 2020." (https://doi.org/10.1007/s10237-020-01307-6)

This paper explores the synergy between exercise and PTH treatment which is essentially a combined treatment. This would be one of the few papers that could be used in the Discussion section as comparison with the current results / model developments.

Sentence containing "(BMD) in specific bone types";

I suggest substituting "types" with "site" as you refer to hip, wrist, vertebra BMD.

---

## [Author Response]

Essential revisions:(1) The discussion needs a lot of revision: it must be clear to authors that the manuscript will be read also by a lot of physicians and a clear, clinical language should be used.

We appreciate the comment as it is our goal to make the manuscript as readable as possible for a large interdisciplinary audience. In response, in the revised manuscript, we have extensively revised the Discussion section, placing special emphasis on the clinical relevance of our findings and avoiding modelling-related jargon as much as possible. Also, in response to point 2, we have added a paragraph on how our findings can be related to fracture risk reduction, the prime clinical goal of osteoporosis therapy (see response below). Since there are many changes which affect the Discussion section as a whole, we refrain from listing each individual change here.

(2) In terms of bone density measurements authors should compare their data with those from Black et al., (JBMR 2018) from the SOF study and Dr Ensrud (JCEM 2022).

We thank the Reviewer for pointing us to the papers by Black et al., (JBMR 33, 389, 2018) and Ensrud et al., (JCEM, *dgac324*, 2022).

First, we would like to recall that our model predicts the time course of the bone mineral density (BMD) for a given drug treatment or ageing scenario. In contrast, the publications by Black et al., and Ensrud et al., report fracture predictions based on BMD measurements. Relating BMD time courses generated by our model to fracture risk would require an additional (statistical) model that depends on details of the target demographic (as does the well-known FRAX tool, which is region-dependent and takes into account a variety of non-BMD related risk factors). This however is a major independent modelling effort and goes beyond the scope of the current approach. However, we believe that the above references (as well as other seminal studies relating BMD and fracture risk) are a perfect occasion to discuss the possibility of creating such a statistical model. In the Discussion section of the revised manuscript, we have therefore included the following new part:

“Clearly, the goal of osteoporosis therapy is the reduction of fracture risk during and after therapy. While BMD has a prime role in the evaluation of osteoporosis therapies and can be measured rather easily using dual-energy x-ray absorptiometry (DXA), its relationship to fracture risk is complex. Fracture risk calculations used in clinical practice also involve demographic and lifestyle-related factors while mostly relying on BMD point measurements (Kanis et al., Bone 44, 2009). However, the quantitative associations between BMD, age and fracture risk reported in many studies (Kanis et al., Osteoporos. Int. 12, 2001; Berger et al., JBMR 24, 2009; Austin et al., JBMR 27, 2012; Krege et al., BoneKEy Rep. 2, 2013; Black et al., JBMR 33, 2018; Ensrud et al., JCEM 2022) can be used to create statistical models that may relate entire BMD time courses to a patient’s fracture risk. Combining such statistical models with the physiology-based model presented here would enable to optimize therapies directly for a minimized long-term fracture risk instead of maximized BMD gain.”

The paper is very well written and presents promising results.The introduction is very comprehensive.

We thank the Reviewer for the positive assessment of our manuscript.

The only additional reference I would suggest is:Lavaill et al., 2021 “Effects of PTH treatment in osteoporosis – insights from a mechanistic PK-PD model, BMMB, pp1-16, 2020.” (https://doi.org/10.1007/s10237-020-01307-6)This paper explores the synergy between exercise and PTH treatment which is essentially a combined treatment. This would be one of the few papers that could be used in the Discussion section as comparison with the current results / model developments.

We thank the Reviewer for pointing us towards this paper. Indeed, physical activity is an important contributor to bone remodelling. As our current model focuses exclusively on pharmacologic treatments and does not capture biomechanical feedback, a direct comparison of results with those by Lavaill et al., is not possible. Nevertheless, the perspective of including the effects of physical activity in an integrated modelling framework is attractive both from a modelling and a therapy standpoint. In the revised discussion, we have therefore added a part highlighting the possible role of physical activity on osteoporosis therapy:

“Physical activity is another important contributor to bone remodeling, which we have not considered here. Detailed modeling approaches involving biomechanical feedback suggest synergistic effects between drug treatment of osteoporosis and physical activity (Lavaill et al., BMMB 19, 2020). Such results call for a further exploration of integrated approaches to osteoporosis treatment combining pharmacologic therapy and lifestyle adjustments.”

We also mention the work in the introduction, where we discuss previous modelling of combination therapies.

Sentence containing "(BMD) in specific bone types";I suggest substituting "types" with "site" as you refer to hip, wrist, vertebra BMD.

We agree with the suggestion and have made the corresponding change in the revised manuscript.

In our initial response to the Public Review, we also promised to describe better the rationale behind the simplified pharmacokinetic description of the various drug classes captured by the model. In the revised manuscript, we have modified and extended the following part in the beginning of Appendix B:

“We include a dynamic description of several drug classes through separate model extensions, which depend on the functional drug concentration. The pharmacodynamic description is drug-specific and represents the individual mechanism of action of the respective drug class. For the pharmacokinetic description of each drug, we resort to simple first-order kinetics with drug-specific half-lives, which reduces the amount of model parameters. More detailed pharmacokinetic descriptions involve drug absorption and transfer between different body compartments, depending on the route of administration (oral, intravenous or subcutaneous). However, simulations of the calibrated model demonstrate that first-order kinetics yields an effective approximation of the pharmacokinetic features essential to capture a drug's long-term effects on bone remodeling, as suggested by comparisons of simulated and measured BTM concentrations (Supplementary Figures 2 and 3).”